# Deep Gaussian Embedding of Graphs: Unsupervised Inductive Learning via Ranking

**Aleksandar Bojchevski, Stephan Günnemann**
Technical University of Munich, Germany
`{a.bojchevski,guennemann}@in.tum.de`

## Abstract

Methods that learn representations of nodes in a graph play a critical role in network analysis since they enable many downstream learning tasks. We propose Graph2Gauss – an approach that can efficiently learn versatile node embeddings on large scale (attributed) graphs that show strong performance on tasks such as link prediction and node classification. Unlike most approaches that represent nodes as point vectors in a low-dimensional continuous space, we embed each node as a Gaussian distribution, allowing us to capture uncertainty about the representation. Furthermore, we propose an unsupervised method that handles inductive learning scenarios and is applicable to different types of graphs: plain/attributed, directed/undirected. By leveraging both the network structure and the associated node attributes, we are able to generalize to unseen nodes without additional training. To learn the embeddings we adopt a personalized ranking formulation w.r.t. the node distances that exploits the natural ordering of the nodes imposed by the network structure. Experiments on real world networks demonstrate the high performance of our approach, outperforming state-of-the-art network embedding methods on several different tasks. Additionally, we demonstrate the benefits of modeling uncertainty – by analyzing it we can estimate neighborhood diversity and detect the intrinsic latent dimensionality of a graph.

## 1 Introduction

Graphs are a natural representation for a wide variety of real-life data, from social and rating networks (Facebook, Amazon), to gene interactions and citation networks (BioGRID, arXiv). Node embeddings are a powerful and increasingly popular approach to analyze such data (Cai et al., 2017). By operating in the embedding space, one can employ proved learning techniques and bypass the difficulty of incorporating the complex node interactions. Tasks such as link prediction, node classification, community detection, and visualization all greatly benefit from these latent node representations. Furthermore, for attributed graphs by leveraging both sources of information (network structure and attributes) one is able to learn more useful representations compared to approaches that only consider the graph (Yang et al., 2015; Pan et al., 2016; Ganguly & Pudi, 2017).

All existing (attributed) graph embedding approaches represent each node by a single point in a low-dimensional continuous vector space. Representing the nodes simply as points, however, has a crucial limitation: we do not have information about the uncertainty of that representation. Yet uncertainty is inherent when describing a node in a complex graph by a single point only. Imagine a node for which the different sources of information are conflicting with each other, e.g. pointing to different communities or even revealing contradicting underlying patterns. Such discrepancy should be reflected in the uncertainty of its embedding. As a solution to this problem, we introduce a novel embedding approach that represents nodes as *Gaussian distributions*: each node becomes a full distribution rather than a single point. Thereby, we capture uncertainty about its representation.

To effectively capture the non-i.i.d. nature of the data arising from the complex interactions between the nodes, we further propose a novel unsupervised *personalized ranking* formulation to learn the embeddings. Intuitively, from the point of view of a single node, we want nodes in its immediate neighborhood to be closest in the embedding space, while nodes multiple hops away should become increasingly more distant. This ordering between the nodes imposed by the network structure w.r.t

the distances between their embeddings naturally leads to our ranking formulation. Taking into account this natural ranking from each node's point of view, we learn more powerful embeddings since we incorporate information about the network structure beyond first and second order proximity.

Furthermore, when node attributes (e.g. text) are available our method is able to leverage them to easily generate embeddings for previously unseen nodes without additional training. In other words, Graph2Gauss is inductive, which is a significant benefit over existing methods that are inherently transductive and do not naturally generalize to unseen nodes. This desirable inductive property comes from the fact that we are learning an encoder that maps the nodes' attributes to embeddings.

The main contributions of our approach are summarized as follows:

a) We embed nodes as Gaussian distributions allowing us to capture **uncertainty**.

b) Our unsupervised **personalized ranking** formulation exploits the natural ordering of the nodes capturing the network structure at multiple scales.

c) We propose an **inductive** method that generalizes to unseen nodes and is applicable to different types of graphs: plain/attributed, directed/undirected.

## 2 RELATED WORK

The focus of this paper is on unsupervised learning of node embeddings for which many different approaches have been proposed. For a comprehensive recent survey see Cai et al. (2017), Hamilton et al. (2017), or Goyal & Ferrara (2017). Approaches such as DeepWalk and node2vec (Perozzi et al., 2014; Grover & Leskovec, 2016) look at plain graphs and learn an embedding based on random walks by extending or adapting the Skip-Gram (Mikolov et al., 2013) architecture. LINE (Tang et al., 2015b) uses first- and second-order proximity and trains the embedding via negative sampling. SDNE (Wang et al., 2016) similarly has a component that preserves second-order proximity and exploits first-order proximity to refine the representations. GraRep (Cao et al., 2015) is a factorization based method that considers local and global structural information.

Tri-Party Deep Network Representation (TRIDNR) (Pan et al., 2016) considers node attributes, network structure and potentially node labels. CENE (Sun et al., 2016) similarly to Ganguly & Pudi (2017) treats the attributes as special kinds of nodes and learns embeddings on the augmented network. Text-Associated DeepWalk (TADW) (Yang et al., 2015) performs low-rank matrix factorization considering graph structure and text features. Heterogeneous networks are consider in (Tang et al., 2015a; Chang et al., 2015), while Huang et al. similarly to Pan et al. (2016) considers labels. GraphSAGE (Hamilton et al., 2017) is an inductive method that generates embeddings by sampling and aggregating attributes from a nodes local neighborhood and requires the edges of the new nodes.

Graph convolutional networks are another family of approaches that adapt conventional CNNs to graph data (Kipf & Welling, 2016a; Defferrard et al., 2016; Henaff et al., 2015; Monti et al., 2016; Niepert et al., 2016; Pham et al., 2017). They utilize the graph Laplacian and the spectral definition of a convolution and boil down to some form of aggregation over neighbors such as averaging. They can be thought of as implicitly learning an embedding, e.g. by taking the output of the last layer before the supervised component. See Monti et al. (2016) for an overview. In contrast to this paper, most of these methods are (semi-)supervised. The graph variational autoencoder (GAE) (Kipf & Welling, 2016b) is a notable exception that learns node embeddings in an unsupervised manner.

Few approaches consider the idea of learning an embedding that is a distribution. Vilnis & McCallum (2014) are the first to learn Gaussian word embeddings to capture uncertainty. Closest to our work, He et al. (2015) represent knowledge graphs and Dos Santos et al. (2016) study heterogeneous graphs for node classification. Both approaches are not applicable for the context of unsupervised learning of (attributed) graphs that we are interested in. The method in He et al. (2015) learns an embedding for each component of the triplets (head, tail, relation) in the knowledge graph. Note that we cannot naively employ this method by considering a single relation "has an edge" and a single entity "node". Since their approach considers similarity between entities and relations, all nodes would be trivially similar to the single relation. Considering the semi-supervised approach proposed in Dos Santos et al. (2016), we cannot simply "turn off" the supervised component to adapt their method for unsupervised learning, since given the defined loss we would trivially map all nodes to the same Gaussian. Additionally, both of these approaches do not consider node attributes.

## 3 Deep Gaussian embedding

In this section we introduce our method Graph2Gauss (G2G) and detail how both the attributes and the network structure influence the learning of node representations. The embedding is carried out in two steps: (i) the node attributes are passed through a non-linear transformation via a deep neural network (encoder) and yield the parameters associated with the node's embedding distribution; (ii) we formulate an unsupervised loss function that incorporates the natural ranking of the nodes as given by the network structure w.r.t. a dissimilarity measure on the embedding distributions.

**Problem definition.** Let $G = (\mathbf{A}, \mathbf{X})$ be a directed attributed graph, where $\mathbf{A} \in \mathbb{R}^{N \times N}$ is an adjacency matrix representing the edges between $N$ nodes and $\mathbf{X} \in \mathbb{R}^{N \times D}$ collects the attribute information for each node where $\mathbf{x}_i$ is a $D$ dimensional attribute vector of the $i^{th}$ node.[1] $V$ denotes the set of all nodes. We aim to find a lower-dimensional Gaussian distribution embedding $\mathbf{h}_i = \mathcal{N}(\mu_i, \Sigma_i)$, $\mu_i \in \mathbb{R}^L, \Sigma_i \in \mathbb{R}^{L \times L}$ with $L \ll N, D$, such that nodes similar w.r.t. attributes and network structure are also similar in the embedding space given a dissimilarity measure $\Delta(\mathbf{h}_i, \mathbf{h}_j)$. In Fig.5(a) for example we show nodes that are embedded as two dimensional Gaussians.

### 3.1 Network structure representation via personalized ranking

To capture the structural information of the network in the embedding space, we propose a personalized ranking approach. That is, locally per node $i$ we impose a ranking of all remaining nodes w.r.t. their distance to node $i$ in the embedding space. More precisely, in this paper we exploit the $k$-hop neighborhoods of each node. Given some anchor node $i$, we define $N_{ik} = \{j \in V | i \neq j, \min(sp(i, j), K) = k\}$ to be the set of nodes who are exactly $k$ hops away from node $i$, where $V$ is the set of all nodes, $K$ is a hyper-parameter denoting the maximum distance we are wiling to consider, and $sp(i, j)$ returns either the length of the shortest path starting at node $i$ and ending in node $j$ or $\infty$ if node $j$ is not reachable.

Intuitively, we want all nodes belonging to the 1-hop neighborhood of $i$ to be closer to $i$ w.r.t. their embedding, compared to the all nodes in its 2-hop neighborhood, which in turn are closer than the nodes in its 3-hop neighborhood and so on up to $K$. Thus, the ranking that we want to ensure from the perspective of node $i$ is

$$\Delta(\mathbf{h}_i, \mathbf{h}_{k_1}) < \Delta(\mathbf{h}_i, \mathbf{h}_{k_2}) < \cdots < \Delta(\mathbf{h}_i, \mathbf{h}_{k_K}) \quad \forall k_1 \in N_{i1}, \forall k_2 \in N_{i2}, \ldots, \forall k_K \in N_{iK}$$

or equivalently, we aim to satisfy the following pairwise constraints

$$\Delta(\mathbf{h}_i, \mathbf{h}_j) < \Delta(\mathbf{h}_i, \mathbf{h}_{j'}), \ \forall i \in V, \ \forall j \in N_{ik}, \ \forall j' \in N_{ik'}, \ \forall k < k'$$

Going beyond mere first-order and second-order proximity this enables us to capture the network structure at multiple scales incorporating local and global structure.

**Dissimilarity measure.** To solve the above ranking task we have to define a suitable dissimilarity measure between the latent representation of two nodes. Since our latent representations are distributions, similarly to Dos Santos et al. (2016) and He et al. (2015) we employ the asymmetric KL divergence. This gives the additional benefit of handling directed graphs in a sound way. More specifically, given the latent Gaussian distribution representation of two nodes $\mathbf{h}_i, \mathbf{h}_j$ we define

$$\Delta(\mathbf{h}_i, \mathbf{h}_j) = D_{KL}(\mathcal{N}_j || \mathcal{N}_i) = \frac{1}{2}\left[ tr(\Sigma_i^{-1} \Sigma_j) + (\mu_i - \mu_j)^T \Sigma_i^{-1} (\mu_i - \mu_j) - L - log\frac{det(\Sigma_j)}{det(\Sigma_i)} \right]$$

Here we use the notation $\mu_i, \Sigma_i$ to denote the outputs of some functions $\mu_\theta(\mathbf{x}_i)$ and $\Sigma_\theta(\mathbf{x}_i)$ applied to the attributes $\mathbf{x}_i$ of node $i$ and $tr(.)$ denotes the trace of a matrix. The asymmetric KL divergence also applies to the case of an undirected graph by simply processing both directions of the edge. We could alternatively use a symmetric dissimilarity measure such as the Jensen-Shannon divergence or the expected likelihood (probability product kernel).

### 3.2 Deep encoder

The functions $\mu_\theta(\mathbf{x}_i)$ and $\Sigma_\theta(\mathbf{x}_i)$ are deep feed-forward non-linear neural networks parametrized by $\theta$. It is important to note that these parameters are shared across instances and thus enjoy statistical

---

[1] Note, in the absence of node attributes we can simply use one-hot encoding for the nodes (i.e. $\mathbf{X} = \mathbf{I}$, where $\mathbf{I}$ is the identity matrix) and/or any other derived features such as node degrees.

strength benefits. Additionally, we design $\mu_\theta(\mathbf{x}_i)$ and $\Sigma_\theta(\mathbf{x}_i)$ such that they share parameters as well. More specifically, a deep encoder $f_\theta(\mathbf{x}_i)$ processes the node's attributes and outputs an intermediate hidden representation, which is then in turn used to output $\mu_i$ and $\Sigma_i$ in the final layer of the architecture. We focus on diagonal covariance matrices.[2] The mapping from the nodes' attributes to their embedding via the deep encoder is precisely what enables the inductiveness of Graph2Gauss.

### 3.3 LEARNING VIA ENERGY-BASED LOSS

Since it is intractable to find a solution that satisfies all of the pairwise constraints defined in Sec. 3.1 we turn to an energy based learning approach. The idea is to define an objective function that penalizes ranking errors given the energy of the pairs. More specifically, denoting the KL divergence between two nodes as the respective energy, $E_{ij} = D_{KL}(\mathcal{N}_j || \mathcal{N}_i)$, we define the following loss to be optimized

$$\mathcal{L} = \sum_i \sum_{k < l} \sum_{j_k \in N_{ik}} \sum_{j_l \in N_{il}} \left( E_{ij_k}{}^2 + \exp^{-E_{ij_l}} \right) = \sum_{(i,j_k,j_l) \in \mathcal{D}_t} \left( E_{ij_k}{}^2 + \exp^{-E_{ij_l}} \right) \quad (1)$$

where $\mathcal{D}_t = \{(i, j_k, j_l) \mid sp(i, j_k) < sp(i, j_l)\}$ is the set of all valid triplets. The $E_{ij_k}$ terms are positive examples whose energy should be lower compared to the energy of the negative examples $E_{ij_l}$. Here, we employed the so called square-exponential loss (LeCun et al., 2006) which unlike other typically used losses (e.g. hinge loss) does not have a fixed margin and pushes the energy of the negative terms to infinity with exponentially decreasing force. In our setting, for a given anchor node $i$, the energy $E_{ij}$ should be lowest for nodes $j$ in his 1-hop neighborhood, followed by a higher energy for nodes in his 2-hop neighborhood and so on.

Finally, we can optimize the parameters $\theta$ of the deep encoder such that the loss $\mathcal{L}$ is minimized and the pairwise rankings are satisfied. Note again that the parameters are shared across all instances, meaning that we share statistical strength and can learn them more easily in comparison to treating the distribution parameters (e.g. $\mu_i$, $\Sigma_i$) independently as free variables. The parameters are optimized using Adam (Kingma & Ba, 2014) with a fixed learning rate of 0.001.

**Sampling strategy.** For large graphs, the complete loss is intractable to compute, confirming the need for a stochastic variant. The naive approach would be to sample triplets from $\mathcal{D}_t$ uniformly, i.e. replace $\sum_{(i,j_k,j_l) \in \mathcal{D}_t}$ with $\mathbb{E}_{(i,j_k,j_l) \sim \mathcal{D}_t}$ in Eq. 1. However, with the naive sampling we are less likely to sample triplets that involve low-degree nodes since high degree nodes occur in many more pairwise constraints. This in turn means that we update the embedding of low-degree nodes less often which is not desirable. Therefore, we propose an alternative node-anchored sampling strategy. Intuitively, for every node $i$, we randomly sample one other node from *each of its neighborhoods* (1-hop, 2-hop, etc.) and then optimize over all the corresponding pairwise constraints $(E_{i1} < E_{i2}, \ldots, E_{i1} < E_{iK}, E_{i2} < E_{i3}, \ldots E_{i2} < E_{iK}, \ldots, E_{iK-1} < E_{iK})$.

Naively applying the node-anchored sampling strategy and optimizing Eq. 1, however, would lead to biased estimates of the gradient. Theorem 1 shows how to adapt the loss such that it is equal in expectation to the original loss under our new sampling strategy. As a consequence, we have unbiased estimates of the gradient using stochastic optimization of the reformulated loss.

**Theorem 1** *For all $i$, let $(j_1, \ldots, j_K)$ be independent uniform random samples from the sets $(N_{i1}, \ldots, N_{iK})$ and $|N_{i*}|$ the cardinality of each set. Then $\mathcal{L}$ is equal in expectation to*

$$\mathcal{L}_s = \sum_i \mathbb{E}_{(j_1,\ldots,j_K) \sim (N_{i1},\ldots,N_{iK})} \left[ \sum_{k<l} |N_{ik}| \cdot |N_{il}| \cdot \left( E_{ij_k}{}^2 + \exp^{-E_{ij_l}} \right) \right] = \mathcal{L} \quad (2)$$

We provide the proof in the appendix. For cases where the number of nodes $N$ is particularly large we can further subsample mini-batches, by selecting anchor nodes $i$ at random. Furthermore, in our experimental study, we analyze the effect of the sampling strategy on convergence, as well as the quality of the stochastic variant w.r.t. the obtained solution and the reached local optima.

---

[2] To ensure that they are positive definite, in the final layer we output $\tilde{\sigma}_{id} \in \mathbb{R}$ and obtain $\sigma_{id} = \text{elu}(\tilde{\sigma}_{id}) + 1$.

### 3.4 DISCUSSION

**Inductive learning.** While during learning we need both the network structure (to evaluate the ranking loss) and the attributes, once the learning concludes, the embedding for a node can be obtained solely based on its attributes. This enables our method to easily handle the issue of obtaining representations for new nodes that were not part of the network during training. To do so we simply pass the attributes of the new node through our learned deep encoder. Most approaches cannot handle this issue at all, with a notable exception being SDNE and GraphSAGE (Wang et al., 2016; Hamilton et al., 2017). However, both approaches require the edges of the new node to get the node's representation, and cannot handle nodes that have no existing connections. In contrast, our method can handle even such nodes, since after the model is learned we rely only on the attribute information.

**Plain graph embedding.** Even though attributed graphs are often found in the real-world, sometimes it is desirable to analyze plain graphs. As already discussed, our method easily handles plain graphs, when the attributes are not available, by using one-hot encoding of the nodes instead. As we later show in the experiments we are able to learn useful representations in this scenario, even outperforming some attributed approaches. Naturally, in this case we lose the inductive ability to handle unseen nodes. We compare the one-hot encoding version, termed G2G_oh, with our full method G2G that utilizes the attributes, as well as all remaining competitors.

**Encoder architecture.** Depending on the type of the node attributes (e.g. images, text) we could in principle use CNNs/RNNs to process them. We could also easily incorporate any of the proposed graph convolutional layers inheriting their benefits. However, we observe that in practice using simple feed-forward architecture with rectifier units is sufficient, while being much faster and easier to train. Better yet, we observed that Graph2Gauss is not sensitive to the choice of hyperparameters such as number and size of hidden layers. We provide more detailed information and sensible defaults in the appendix.

**Complexity.** The time complexity for computing the original loss is $O(N^3)$ where $N$ is the number of nodes. Using our node-anchored sampling strategy, the complexity of the stochastic version is $O(K^2N)$ where $K$ is the maximum distance considered. Since a small value of $K \leq 2$ consistently showed good performance, $K^2$ becomes negligible and thus the complexity is $O(N)$, meaning linear in the number of nodes. This coupled with the small number of epochs $T$ needed for convergence ($T \leq 2000$ for all shown experiments, see e.g. Fig. 3(b)) and an efficient GPU implementation also made our method faster than most competitors in terms of wall-clock time.

## 4 EMBEDDING EVALUATION

We compare Graph2Gauss with and without considering attributes (G2G, G2G_oh) to several competitors namely: TRIDNR and TADW (Pan et al., 2016; Yang et al., 2015) as representatives that consider attributed graphs, GAE (Kipf & Welling, 2016b) as the unsupervised graph convolutional representative, and node2vec (Grover & Leskovec, 2016) as a representative of the random walk based plain graph embeddings. Additionally, we include a strong Logistic Regression baseline that considers only the attributes. As with all other methods we train TRIDNR in a unsupervised manner, however, since it can only process raw text as attributes (rather than e.g. bag-of-words) it is not always applicable. Furthermore, since TADW, and GAE only support undirected graphs we must symmetrize the graph before using them – giving them a substantial advantage, especially in the link prediction task. Moreover, in all experiments if the competing techniques use an $L$ dimensional embedding, G2G's embedding is actually only *half* of this dimensionality so that the overall number of 'parameters' per node (mean vector + variance terms of the diagonal $\Sigma_i$) matches $L$.

**Dataset description.** We use several attributed graph datasets. Cora (McCallum et al., 2000) is a well-known citation network labeled based on the paper topic. While most approaches report on a small subset of this dataset we additionally extract from the original data the entire network and name these two datasets **CORA** ($N = 19793, E = 65311, D = 8710, K = 70$) and **CORA-ML** ($N = 2995, E = 8416, D = 2879, K = 7$) respectively. **CITESEER** ($N = 4230, E = 5358, D = 2701, K = 6$) (Giles et al., 1998), **DBLP** (Pan et al., 2016) ($N = 17716, E = 105734, D = 1639, K = 4$) and **PUBMED** ($N = 18230, E = 79612, D = 500, K = 3$) (Sen et al., 2008) are other commonly used citation datasets. We provide all datasets, the source code of G2G, and further supplementary material (https://www.kdd.in.tum.de/g2g).

## 4.1 LINK PREDICTION

**Setup.** Link prediction is a commonly used task to demonstrate the meaningfulness of the embeddings. To evaluate the performance we hide a set of edges/non-edges from the original graph and train on the resulting graph. Similarly to Kipf & Welling (2016b) and Wang et al. (2016) we create a validation/test set that contains $5\%/10\%$ randomly selected edges respectively and equal number of randomly selected non-edges.We used the validation set for hyper-parameter tuning and early stopping and the test set only to report the performance. As by convention we report the area under the ROC curve (AUC) and the average precision (AP) scores for each method. To rank the candidate edges we use the negative energy $-E_{ij}$ for Graph2Gauss, and the exact same approach as in the respective original methods (e.g. dot product of the embeddings).

**Performance on real-world datasets.** Table 1 shows the performance on the link prediction task for different datasets and embedding size $L = 128$. As we can see our method significantly outperforms the competitors across all datasets which is a strong sign that the learned embeddings are useful. Furthermore, even the constrained version of our method G2G_oh that does not consider attributes at all outperforms the competitors on some datasets. While GAE achieves comparable performance on some of the datasets their approach doesn't scale to large graphs. In fact, for graphs beyond $15K$ nodes we had to revert to slow training on the CPU since the data did not fit on the GPU memory (12GB). The simple Logistic Regression baseline showed surprisingly strong performance, even outperforming some of the more complicated methods.

Table 1: Link prediction performance for real-world datasets with $L = 128$.

| Method | Cora-ML | | Cora | | Citeseer | | DBLP | | Pubmed | | Cora-ML Easy | |
|---|---|---|---|---|---|---|---|---|---|---|---|---|
| | AUC | AP | AUC | AP | AUC | AP | AUC | AP | AUC | AP | AUC | AP |
| Logistic Regression | 90.01 | 89.75 | 86.58 | 86.51 | 81.70 | 79.10 | 82.04 | 81.91 | 90.50 | 90.99 | 90.28 | 90.99 |
| node2vec(Grover & Leskovec, 2016) | 76.80 | 75.26 | 79.95 | 78.98 | 83.04 | 83.74 | 95.42 | 95.33 | 95.42 | 95.33 | 93.47 | 93.53 |
| TADW(Yang et al., 2015) | 81.26 | 81.34 | 76.56 | 78.06 | 70.14 | 72.93 | 65.67 | 59.85 | 62.72 | 68.02 | 83.53 | 82.47 |
| TRIDNR(Pan et al., 2016) | 84.51 | 85.69 | 81.61 | 81.08 | 87.23 | 88.87 | 92.01 | 91.62 | NTA | NTA | 85.59 | 86.16 |
| GAE(Kipf & Welling, 2016b) | 96.65 | 96.67 | 97.91 | 98.07 | 92.31 | 93.88 | 95.78 | 96.67 | 96.07 | 96.12 | 95.97 | 95.17 |
| G2G_oh | 96.95 | 97.54 | 98.41 | 98.63 | 95.89 | 95.78 | 98.29 | 98.46 | 96.75 | 96.47 | 96.98 | 96.42 |
| G2G | **98.01** | **98.03** | **98.81** | **98.78** | **96.09** | **96.16** | **98.65** | **98.78** | **97.42** | **97.85** | **98.03** | **98.12** |

We also include the performance on the so called "Cora-ML Easy" dataset, obtained from the Cora-ML dataset by making it undirected and selecting the nodes in the largest connected component. We see that while node2vec struggles on the original real-world data, it significantly improves in this "easy" setting. On the contrary, Graph2Gauss handles both settings effortlessly. This demonstrates that Graph2Gauss can be readily applied in realistic scenarios on potentially messy real-world data.

**Sensitivity analysis.** In Figs.1(a) and 1(b) we show the performance w.r.t. the dimensionality of the embedding, averaged over 10 trials. G2G is able to learn useful embeddings with strong performance even for relatively small embedding sizes. Even for the case $L = 2$, where we embed the points as one dimensional Gaussian distributions ($L = 1 + 1$ for the mean and the sigma of the Gaussian), G2G still outperforms all of the competitors irrespective of their much higher embedding sizes.

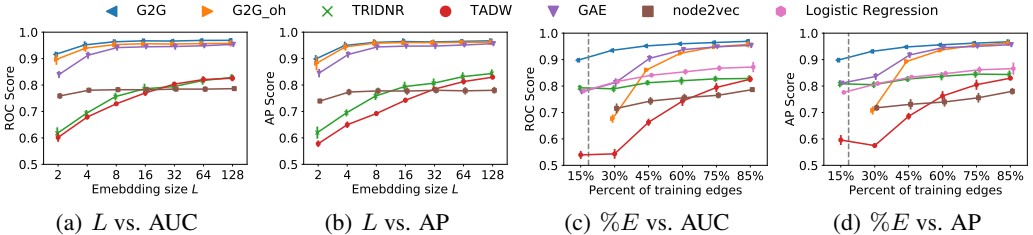

(a) $L$ vs. AUC      (b) $L$ vs. AP      (c) $\%E$ vs. AUC      (d) $\%E$ vs. AP

Figure 1: Link prediction performance for different embedding sizes and percentages of training edges on Cora-ML. G2G outperforms the competitors even for small sizes and percentage of edges.

Finally, we evaluate the performance w.r.t. the percentage of training edges varying from $15\%$ to $85\%$, averaged over 10 trials. We can see in Figs.1(c) and 1(d) Graph2Gauss strongly outperforms the competitors, especially for small number of training edges. The dashed line indicates the percent-

age above which we can guarantee to have every node appear at least once in the training set.[3] The performance below that line is then indicative of the performance in the inductive setting. Since, the structure only methods are unable to compute meaningful embeddings for unseen nodes we cannot report their performance below the dashed line.

## 4.2 NODE CLASSIFICATION

**Setup.** Node classification is another task commonly used to evaluate the strength of the learned embeddings – after they have been trained in an unsupervised manner. We evaluate the node classification performance for three datasets (Cora-ML, Citeseer and DBLP) that have ground-truth classes. First, we train the embeddings on the entire training data in an unsupervised manner (excluding the class labels). Then, following Perozzi et al. (2014) we use varying percentage of randomly selected nodes and their learned embeddings along with their labels as training data for a logistic regression, while evaluating the performance on the rest of the nodes. We also optimize the regularization strength for each method/dataset via cross-validation. We show results averaged over 10 trials.

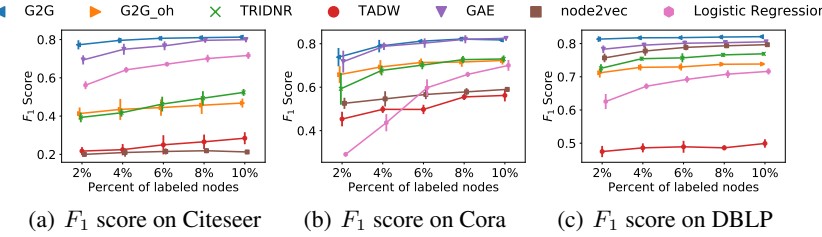

(a) $F_1$ score on Citeseer     (b) $F_1$ score on Cora     (c) $F_1$ score on DBLP

Figure 2: Classification performance comparison - both G2G and G2G_oh perform strongly.

**Performance on real-world datasets.** Figs. 2 compares the methods w.r.t. the classification performance for different percentage of labeled nodes. We can see that our method clearly outperforms the competitors. Again, the constrained version of our method that does not consider attributes is able to outperform some of the competing approaches. Additionally, we can conclude that in general our method shows stable performance regardless of the percentage of labeled nodes. This is a highly desirable property since it shows that should we need to perform classification it is sufficient to train only on a small percentage of labeled nodes.

## 4.3 SAMPLING STRATEGY

Figure 3(a) shows the validation set ROC score for the link prediction task w.r.t. the number of triplets $(i, j_k, j_l)$ seen. We can see that both sampling strategies are able to reach the same performance as the full loss in significantly fewer ($< 4.2\%$) number of pairs seen (note the log scale). It also shows that the naive random sampling converges slower than the node-anchored sampling strategy. Figures 3(b) gives us some insight as to why – our node-anchored sampling strategy achieves significantly lower loss. Finally, Fig. 3(c) shows that our node-anchored sampling strategy has lower variance of the gradient updates, which is another contributor to faster convergence.

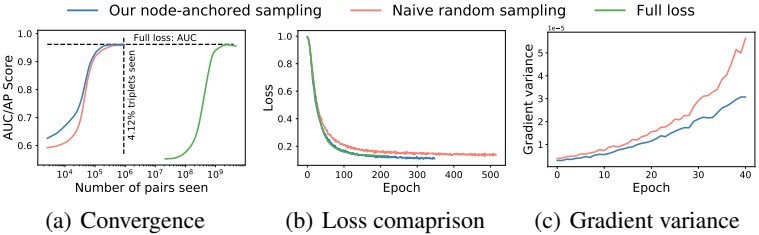

(a) Convergence     (b) Loss comaprison     (c) Gradient variance

Figure 3: Our sampling strategy converges significantly faster than the full loss, while maintaining good performance. It also achieves better loss and has lower variance compared to naive sampling.

---

[3]This percentage is derived from the size of the minimum edge-cover set. For more details see appendix.

## 4.4 EMBEDDING UNCERTAINTY

Learning an embedding that is a distribution rather than a point-vector allows us to capture uncertainty about the representation. We perform several experiments to evaluate the benefit of modeling uncertainty. Figure 4(a) shows that the learned uncertainty is correlated with neighborhood diversity, where for a node $i$ we define diversity as the number of distinct classes among the nodes in its $p$-hop neighborhood ($\bigcup_{1 \leq k \leq p} N_{ik}$). Since the uncertainty for a node $i$ is an $L$-dimensional vector (diagonal covariance) we show the average across the dimensions. In line with our intuition, nodes with less diverse neighborhood have significantly lower variance compare to more diverse nodes whose immediate neighbors belong to many different classes, thus making their embedding more uncertain. The figure shows the result on the Cora dataset for $p = 3$ hop neighborhood. Similar results hold for the other datasets. This result is particularly impressive given the fact that we learn our embedding in a completely unsupervised manner, yet the uncertainty was able to capture the diversity w.r.t. the class labels of the neighbors of a node, which were never seen during training.

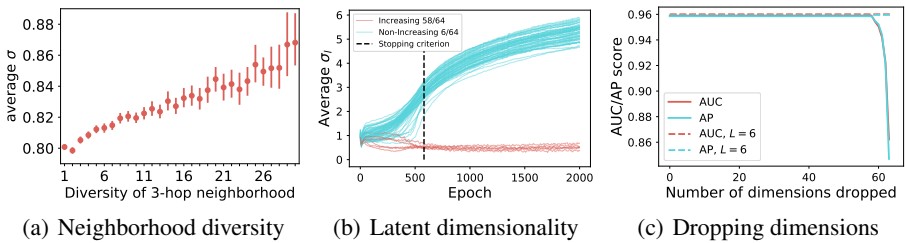

(a) Neighborhood diversity    (b) Latent dimensionality    (c) Dropping dimensions

Figure 4: The benefit of modeling the uncertainty of the nodes.

Figure 4(b) shows that using the learned uncertainty we are able to detect the intrinsic latent dimensionality of the graph. Each line represents the average variance (over all nodes) for a given dimension $l$ for each epoch. We can see that as the training progresses past the stopping criterion (link prediction performance on validation set) and we start to overfit, some dimensions exhibit a relatively stable average variance, while for others the variance increases with each epoch. By creating a simply rule that monitors the average change of the variance over time we were able to automatically detect these relevant latent dimensions (colored in red). This result holds for multiple datasets and is shown here for Cora-ML. Interestingly, the number of detected latent dimensions (6) is close to the number of ground-truth communities (7).

The next obvious question is then how does the performance change if we remove these highly uncertain dimensions whose variance keeps increasing with training. Figure 4(c) answers exactly that. By removing progressively more and more dimensions, starting with the most uncertain first we see imperceptibly small change in performance. Only once we start removing the true latent dimension we see a noticeable degradation in performance. The dashed lines show the performance if we re-train the model, setting $L = 6$, equal to the detected number of latent dimensions.

As a last study of uncertainty, in a use case analysis, the nodes with high uncertainty reveal additional interesting patterns. For example in the Cora dataset, one of the highly uncertain nodes was the paper "The use of word shape information for cursive script recognition" by R.J. Whitrow – surprisingly, all citations (edges) of that paper (as extracted from the dataset) were towards other papers by the same author.

## 4.5 INDUCTIVE LEARNING: GENERALIZATION TO UNSEEN NODES

As discussed in Sec. 3.4 G2G is able to learn embeddings even for nodes that were not part of the networks structure during training time. Thus, it not only supports transductive but also inductive learning. To evaluate how our approach generalizes to unseen nodes we perform the following experiment: (i) first we completely hide $10\%/25\%$ of nodes from the network at random; (ii) we proceed to learn the node embeddings for the rest of the nodes; (iii) after learning is complete we pass the (new) unseen test nodes through our deep encoder to obtain their embedding; (iv) we evaluate by calculating the link prediction performance (AUC and AP scores) using all their edges and same number of non-edges.

Table 2: Inductive link prediction performance.

| Method (% hidden) | Cora-ML | | Cora | | Citeseer | | DBLP | | Pubmed | |
|---|---|---|---|---|---|---|---|---|---|---|
| | AUC | AP | AUC | AP | AUC | AP | AUC | AP | AUC | AP |
| Log.Reg. 10% | 75.95 | 78.62 | 78.53 | 78.70 | 73.09 | 72.54 | 67.55 | 69.55 | 86.83 | 87.34 |
| G2G 10% | 90.93 | 89.37 | 94.18 | 93.40 | 88.58 | 88.31 | 85.06 | 83.75 | 92.22 | 90.45 |
| G2G 25% | 87.83 | 86.31 | 92.96 | 92.31 | 87.30 | 86.61 | 83.09 | 81.49 | 90.20 | 88.28 |

As the results in Table 2 clearly show, since we are utilizing the rich attribute information, we are able to achieve strong performance for unseen nodes. This is true even when a quarter of the nodes are missing. This makes our method applicable in the context of large graphs where training on the entire network is not feasible. Note that SDNE (Wang et al., 2016) and GraphSAGE (Hamilton et al., 2017) cannot be applied in this scenario, since they also require the edges for the unseen nodes to produce an embedding. Graph2Gauss is the only inductive method that can obtain embeddings for a node based *only* on the node attributes.

## 4.6 NETWORK VISUALIZATION

One key application of node embedding approaches is creating meaningful visualizations of a network in 2D/3D that support tasks such as data exploration and understanding. Following Tang et al. (2015b) and Pan et al. (2016) we first learn a lower-dimensional $L = 128$ embedding for each node and then map those representations in 2D with TSNE (Maaten & Hinton, 2008). Additionally, since our method is able to learn useful representations even in low dimensions we embed the nodes as 2D Gaussians and visualize the resulting embedding. This has the added benefit of visualizing the nodes' uncertainty as well. Fig. 5 shows the visualization for the Cora-ML dataset. We see that Graph2Gauss learns an embedding in which the different classes are clearly separated.

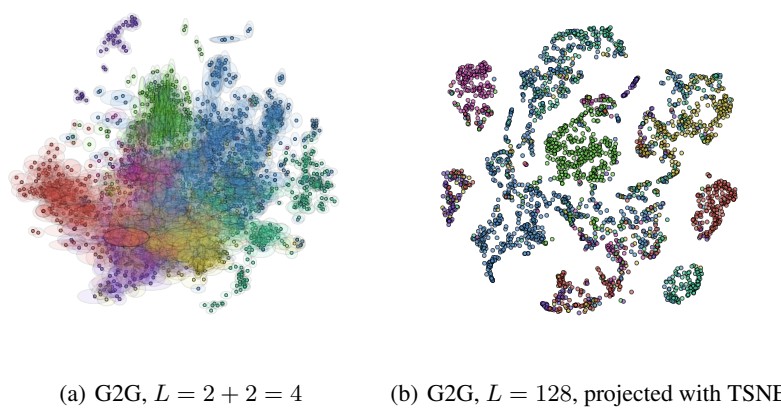

(a) G2G, $L = 2 + 2 = 4$      (b) G2G, $L = 128$, projected with TSNE

Figure 5: 2D visualization of the embeddings on the Cora-ML dataset. Color indicates the class label not used during training. Best viewed on screen.

## 5 CONCLUSION

We proposed Graph2Gauss – the first unsupervised approach that represents nodes in attributed graphs as Gaussian distributions and is therefore able to capture uncertainty. Analyzing the uncertainty reveals the latent dimensionality of a graph and gives insight into the neighborhood diversity of a node. Since we exploit the attribute information of the nodes we can effortlessly generalize to unseen nodes, enabling inductive reasoning. Graph2Gauss leverages the natural ordering of the nodes w.r.t. their neighborhoods via a personalized ranking formulation. The strength of the learned embeddings has been demonstrated on several tasks – specifically achieving high link prediction performance even in the case of low dimensional embeddings. As future work we aim to study personalized rankings beyond the ones imposed by the shortest path distance.

## ACKNOWLEDGMENTS

This research was supported by the German Research Foundation, Emmy Noether grant GU 1409/2-1, and by the Technical University of Munich - Institute for Advanced Study, funded by the German Excellence Initiative and the European Union Seventh Framework Programme under grant agreement no 291763, co-funded by the European Union.

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

APPENDIX

## A  PROOF OF THEOREM 1

To prove Theorem 1 we start with the loss $\mathcal{L}_s$ (Eq. 2), and show that by applying the expectation operator we will obtain the original loss $\mathcal{L}$ (Eq. 1). From there it trivially follows that taking the gradient with respect to $\mathcal{L}_s$ for a set of samples gives us an unbiased estimate of the gradient of $\mathcal{L}$.

First we notice that both $\mathcal{L}$ and $\mathcal{L}_s$ are summing over $i$, thus it is sufficient to show that the losses are equal in expectation for a single node $i$. Denoting with $\mathcal{L}_s^{(i)}$ the loss for a single node $i$ and with $E_{i,k,l} = E_{ij_k}{}^2 + \exp^{-E_{ij_l}}$ for notational convenience we have:

$$
\begin{aligned}
\mathcal{L}_s^{(i)} =& \mathbb{E}_{(j_1,\ldots,j_K)\sim(N_{i1},\ldots,N_{iK})} \sum_{k<l} |N_{ik}| \cdot |N_{il}| \cdot E_{i,k,l} \\
\overset{(1)}{=}& \mathbb{E}_{(j_1,\ldots,j_K)\sim(N_{i1},\ldots,N_{iK})} |N_{i1}| \cdot |N_{i2}| \cdot E_{i,1,2} \\
&+ \cdots + \mathbb{E}_{(j_1,\ldots,j_K)\sim(N_{i1},\ldots,N_{iK})} |N_{iK-1}| \cdot |N_{iK}| \cdot E_{i,K-1,K} \\
\overset{(2)}{=}& \mathbb{E}_{(j_1,j_2)\sim(N_{i1},N_{i2})} |N_{i1}| \cdot |N_{i2}| \cdot E_{i,1,2} \\
&+ \cdots + \mathbb{E}_{(j_{K-1},j_K)\sim(N_{iK-1},N_{iK})} |N_{iK-1}| \cdot |N_{iK}| \cdot E_{i,K-1,K} \\
\overset{(3)}{=}& \sum_{j_1 \in N_{i1}} \sum_{j_2 \in N_{i2}} p(j_1)p(j_2) |N_{i1}| \cdot |N_{i2}| \cdot E_{i,1,2} \\
&+ \cdots + \sum_{j_{K-1} \in N_{iK-1}} \sum_{j_K \in N_{iK}} p(j_{K-1})p(j_K) |N_{iK-1}| \cdot |N_{iK}| \cdot E_{i,K-1,K} \\
\overset{(4)}{=}& \frac{1}{|N_{i1}|} \frac{1}{|N_{i2}|} |N_{i1}||N_{i2}| \sum_{j_1 \in N_{i1}} \sum_{j_2 \in N_{i2}} \cdot E_{i,1,2} \\
&+ \cdots + \frac{1}{|N_{iK-1}|} \frac{1}{|N_{iK}|} |N_{iK-1}||N_{iK}| \sum_{j_{K-1} \in N_{iK-1}} \sum_{j_K \in N_{iK}} \cdot E_{i,K-1,K} \\
=& \sum_{j_1 \in N_{i1}} \sum_{j_2 \in N_{i2}} \cdot E_{i,1,2} + \cdots + \sum_{j_{K-1} \in N_{iK-1}} \sum_{j_K \in N_{iK}} \cdot E_{i,K-1,K} \\
=& \sum_{k<k'} \sum_{j \in N_{ik}} \sum_{j' \in N_{ik'}} \left( E_{ij}{}^2 + \beta \cdot \exp^{-E_{ij'}} \right)
\end{aligned}
$$

In step (1) we have expanded the sum over $k < l$ in independent terms. In step (2) we have marginalized the expectation over the variables that do not appear in the expression, e.g. for the term $\mathbb{E}_{(j_1,\ldots,j_K)\sim(N_{i1},\ldots,N_{iK})} |N_{i1}| \cdot |N_{i2}| \cdot E_{i12}$ we can marginalize over $j_p$ where $p \neq 1$ and $p \neq 2$ since the term doesn't depend on them. In step (3) we have expanded the expectation term. In step (4) we have substituted $p(j_p)$ with $\frac{1}{|N_{ij_p}|}$ since we are sampling uniformly at random.

Since $\mathcal{L}_s^{(i)}$ is equal to $\mathcal{L}^{(i)}$ in expectation it follows that $\nabla \mathcal{L}_s$ based on a set of samples is an unbiased estimate of $\nabla \mathcal{L}$.

## B  IMPLEMENTATION DETAILS

**Architecture and hyperparameters.** We observed that Graph2Gauss is not sensitive to the choice of hyperparameters such as number and size of hidden layers. Better yet, as shown in Sec. 4.4, Graphs2Gauss is also not sensitive to the size of the embedding $L$. Thus, for a new graph, one can simply pick a relatively large embedding size and if required prune it later similarly to the analysis performed in Fig. 4(c).

As a sensible default we recommend an encoder with a single hidden layer of size $s_1 = 512$. More specifically, to obtain the embeddings for a node $i$ we have

$$\mathbf{h}_i = \text{relu}(\mathbf{X}_i \mathbf{W} + \mathbf{b}) \qquad \mu_i = \mathbf{h_i} \mathbf{W}_\mu + \mathbf{b}_\mu \qquad \sigma_i = \text{elu}(\mathbf{h}_i \mathbf{W}_\mathbf{\Sigma} + \mathbf{b}_\mathbf{\Sigma}) + 1$$

where $\mathbf{x}_i$ are node attributes, relu and elu are the rectified linear unit and exponential linear unit respectively. In practice, we found that the softplus works equally well as the elu for making sure that $\sigma_i$ are positive and in turn $\Sigma_i$ is positive definite. We used Xavier initialization (Glorot & Bengio, 2010) for the weight matrices $\mathbf{W} \in \mathbb{R}^{D \times s_1}$, $\mathbf{b} \in \mathbb{R}^{s_1}$, $\mathbf{W}_\mu \in \mathbb{R}^{s_1 \times L/2}$, $\mathbf{b}_\mu \in \mathbb{R}^{L/2}$, $\mathbf{W}_\mathbf{\Sigma} \in \mathbb{R}^{s_1 \times L/2}$, $\mathbf{b}_\mathbf{\Sigma} \in \mathbb{R}^{L/2}$. As discussed in Sec. 3.4, multiple hidden layers, or other architectures such as CNNs/RNNs can also be used based on the specific problem.

Unlike other approaches using Gaussian embeddings (Vilnis & McCallum, 2014; He et al., 2015; Dos Santos et al., 2016) we do not explicitly regularize the norm of the means and we do not clip the covariance matrices. Given the self-regularizing nature of the KL divergence this is unnecessary, as was confirmed in our experiments. The parameters are optimized using Adam (Kingma & Ba, 2014) with a fixed learning rate of $0.001$ and no learning rate annealing/decay.

**Edge cover.** Some of the methods such as node2vec (Grover & Leskovec, 2016) are not able to produce an embedding for nodes that have not been seen during training. Therefore, it is important to make sure that during the train-validation-test split of the edge set, every node appears at least once in the train set. Random sampling of the edges does not guarantee this, especially when allocating a low percentage of edges in the train set during the split. To guarantee that every node appears at least once in the train set we have to find an edge cover. An edge cover of a graph is a set of edges such that every node of the graph is incident to at least one edge of the set. The minimum edge cover problem is the problem of finding an edge cover of minimum size. The dashed line in Figures 1(c) and 1(d) indicates exactly the size of the minimum edge cover. This condition had to be satisfied for the competing methods, however, since Graph2Gauss is inductive, it does not require that every node is in the train set.

