# OpenReview forum: "Deep Gaussian Embedding of Graphs: Unsupervised Inductive Learning via Ranking"
_ICLR.cc/2018/Conference — Accept (Poster)_

### Official Review · AnonReviewer3 · 2017-11-26
**This work proposed a method to embed a network with node attributes into multi-dimensional Gaussian distributions that captures uncertainties and is capable of generalizing to unseen nodes without further training. The model adopts a neural network to transform node attributes into parameters for embedding distributions.**

**Rating:** 7
**Confidence:** 4

**Review:**

This paper is well-written and easy follow. I didn't find serious concern and therefore suggest an acceptance.

Pros
Methodology
1. inductive ability: can generalize to unseen nodes without any further training
2. personalized ranking: the model uses natural ranking that embeddings of closer nodes (considers node pairs of any distance) should be closer in the embedding space, which is more general than prevailing first and second order proximity
3. sampling strategy: the proposed node-anchored sampling method gives unbiased estimates of loss function and successfully reduces the time complexity

Experiment
1. Evaluation tasks including link prediction and node classification are conducted across multiple datasets with additional parameter sensitivity and missing-link robustness experiments
2. Compared with various baselines with diverse model designs such as GCN and node2vec as well as compared with naive baseline (using original node attributes as model inputs)
3. Demonstrated the model captures uncertainties and the learned uncertainties can be used to infer latent dimensions
Related Works
The survey of related work is sufficiently wide and complete.

Cons
Authors should include which kind of model is used to do the link prediction task given embedding vectors from different models as inputs.

---

> ### Author Response · Authors · 2017-12-04
> **Re: This work proposed a method to embed a network with node attributes into multi-dimensional Gaussian distributions...**
>
> Thank you for your review and comments.
>
> Regarding the model used to do the link prediction task we adopt the exact same approach as described in the respective original methods of each of the competitors (e.g we use the dot product of the embeddings). For Graph2Gauss the negative energy (-E_ij) is used for ranking candidate links. Note that the two metrics AUC and AP do not need a binary decision (edge/non-edge), but rather a (possibly unnormalized) score indicating how likely is the edge. We now include these details in the uploaded revised version.

---

### Official Review · AnonReviewer2 · 2017-11-28
**The paper brings some new ideas for learning unsupervised graph node embeddings. The experiments are fine, but not so conclusive.**

**Rating:** 6
**Confidence:** 4

**Review:**

The paper proposes to learn Gaussian embeddings for directed attributed graph nodes. Each node is associated to a Gaussian representation (mean and diagonal covariance matrix). The mean and diagonal representations for a node are learned as functions of the node attributes. The algorithm is unsupervised and optimizes a ranking loss: nodes at distance 1 in the graph are closer than nodes at distance 2, etc. Distance between nodes representation is measured via KL divergence. The ranking loss is a square exponential loss proposed in energy based models. In order to limit the complexity, the authors propose the use of a sampling scheme and show the convergence in expectation of this strategy towards the initial loss. Experiments are performed on two tasks: link prediction and node classification. Baselines are unsupervised projection methods and a (supervised) logistic regression. An analysis of the algorithm behavior is then proposed.
The paper reads well. Using a ranking loss based on the node distance together with Gaussian embeddings is probably new, even if the novelty is not that big.  The comparisons with unsupervised methods shows that the algorithm learns relevant representations.
Do you have a motivation for using this specific loss Eq. (1), or is it a simple heuristic choice? Did you try other ranking losses?
For the link prediction experiments, it is not indicated how you rank candidate links for the different methods and how you proceed with the logistic. Did you compare with a more complex supervised model than the logistic? Fort the classification tasks, it would be interesting to compare to supervised/ semi-supervised embedding methods. The performance of unsupervised embeddings for graph node classification is usually much lower than supervised/ semi-supervised methods.  Having a measure of the performance gap on the different tasks would be informative. Concerning the analysis of uncertainity, discovering that uncertainty is higher for nodes with neighbors of distinct classes is interesting. In your setting this might simply be caused by the difference in the node attributes. I was not so convinced by the conclusions on the dimensionality of the hidden representation space. An immediate conclusion of this experiment would be that only a small dimensional latent space is needed. Did you experiment with this?
Detailed comments:
The title of the paper is “Deep …”. There is nothing Deep in the proposed model since the NN are simple one layer MLPs. This is not a criticism, but the title should be changed.
There is a typo in KL definition (d should be replaced by the dimension of the embeddings). Probably another typo: the energy should be + D_KL and not –D_KL. The paragraph below eq (1) should be modified accordingly.
All the figures are too small to see anything and should be enlarged.
Overall the paper brings some new ideas. The experiments are fine, but not so conclusive.

---

> ### Author Response · Authors · 2017-12-04
> **Re: The paper brings some new ideas for learning unsupervised graph node embeddings...**
>
> Thank you for your review and comments. We provide answers to all your questions.
>
> 1) Eq. (1) motivation:
> Three types of loss functions are typically considered in the ranking literature: pointwise, pairwise and listwise. We employ the pairwise approach since it usually outperforms the pointwise approach and compared to the listwise approach it is more amenable to stochastic training. The listwise approach is also computationally more expensive and early experiments did not show any benefits of using it. Regarding the pairwise loss function we indeed considered several forms typically used in energy-based learning, including the square-exponential, the hinge loss, LVQ2 and others. They performed comparatively. The final choice of the square-exponential is because compared to e.g. the hinge loss and LVQ2 we don't have the need for tuning a hyperparameter such as the margin.
>
> 2) Link prediction:
> Note that the two metrics AUC and AP do not need a binary decision (edge/non-edge), but rather a (possibly unnormalized) score indicating how likely is the edge. To rank candidate links (i.e. obtain the score) we adopt the exact same approach as described in the respective original methods of each of the competitors (e.g we use the dot product of the embeddings). For Graph2Gauss the negative energy (-E_ij) is used for ranking candidate links. We now include these details in the uploaded revised version.
>
> 3) Logistic regression:
> We used the logistic regression as a supervised model since this is a common choice used in almost all previous node embedding papers.
>
> 4) Supervised/semi-supervised method:
> It is expected that the performance of supervised/semi-supervised method would be stronger, especially on the node classification task. However, as we already state in the related work section the focus of this paper is on unsupervised learning. While additional comparison with different supervised/semi-supervised methods would be beneficial, we feel this would distract the reader from the main goal: "unsupervised learning of node embeddings". Furthermore, it would be straightforward to extend Graph2Gauss to the semi-supervised setting by including a supervised component in the loss, and we leave this for future work.
>
> 5) Uncertainty/dimensionality:
> The conclusion that only a small dimensional latent space is needed is correct and we indeed experimented with this: the sensitivity analysis in Figures 1a) and 1b) shows that increasing the latent dimensionality beyond some small (dataset-specific) number doesn't give significant increase in performance and the performance flattens out. The benefit of the uncertainty analysis is that for a new dataset we would not need to do train multiple models with different latent dimensions such as in Figures 1a) and 1b) to determine what is the minimum number of dimensions for good performance. We could instead train the model with a large latent dimension and perform analysis similar to the one in Figure 4c).
>
> 6) Deep:
> We used "Deep" in the title, since the general architecture is conceived with multiple layers in mind. However, in our experiments single hidden layers proved to be enough to reach good performance. We could certainly change the the title to reflect this.
>
> 7) KL/Energy typo:
> It is true, the KL definition and energy have a typo. We have already fixed this in the uploaded revised version. This question was asked earlier and we also answer it in more details in the comment below.
>
> 8) Readability:
> We agree that readability is important and we and we will enlarge the figures.

---

### Official Review · AnonReviewer1 · 2017-12-03
**Deep Gaussian Embedding of Graphs: Unsupervised Inductive Learning via Ranking**

**Rating:** 7
**Confidence:** 3

**Review:**

This paper proposes Graph2Gauss (G2G), a node embedding method that embeds nodes in attributed graphs (can work w/o attributes as well) into Gaussian distributions rather than conventionally latent vectors. By doing so, G2G can reflect the uncertainty of a node's embedding. The authors then use these Gaussian distributions and neighborhood ranking constraints to obtain the final node embeddings. Experiments on link prediction and node classification showed improved performance over several strong embedding methods. Overall, the paper is well-written and the contributions are remarkable. The reason I am giving a less possible rating is that some statements are questionable and can severely affect the conclusions claimed in this paper, which therefore requires the authors' detailed response. I am certainly willing to change my rating if the authors clarify my questions.

Major concern 1: Is the latent vector dimension L really the same for G2G and other compared methods?
In the first paragraph of Section 4, it is stated that "in all experiments if the competing techniques use an embedding of
dimensionality L, G2G’s embedding is actually only half of this dimensionality so that the overall number of ’parameters’ per node (mean vector + variance terms) matches L."  This setting can be wrong since the degree of freedom of a L-dim Gaussian distribution should be L+L(L-1)/2, where the first term corresponds to the mean and the second term corresponds to the covariance. If I understand it correctly, when any compared embedding method used an L-dim vector, the authors used the dimension of L/2. But this setting is wrong if one wants the overall number of ’parameters’ per node (mean vector + variance terms) matches L, as stated by the authors. Fixing L, the equivalent dimension L_G2G for G2G should be set such that L_G2G +L_G2G (L_G2G -1)/2=L, not 2*L_G2G=L.  Since this setting is universal to the follow-up analysis and may severely degrade the performance of GSG due to less embedding dimensions, I hope the authors can clarify this point.

Major concern 2: The claim on inductive learning
Inductive learning is one of the major contributions claimed in this paper. The authors claim G2G can learn an embedding of an unseen node solely based on their attributes. However, is it not clear why this can be done. In the learning stage of Sec. 3.3, the attributes do not seem to play a role in the energy function. Also, since no algorithm descriptions are available, it's not clear how using only an unseen node's attributes can yield a good embedding under G2G work (so does Sec. 4.5).
Moreover, how does it compare to directly using raw user attributes for these tasks?

Minor concern/suggestions: The "similarity" measure in section 3.1 using KL divergence should be better rephased by "dissimilarity" measure. Otherwise, one has a similarity measure $Delta$ and wants it to increase as the hop distance k decreases (closer nodes are more similar). But the ranking constraints are somewhat counter-intuitive because you want $Delta$ to be small if nodes are closer. There is nothing wrong with the ranking condition, but rather an inconsistency between the use of "similarity" measure for KL divergence.

---

> ### Author Response · Authors · 2017-12-04
> **Re: Deep Gaussian Embedding of Graphs: Unsupervised Inductive Learning via Ranking**
>
> Thank you for your review and comments. We provide clarification for all your concerns.
>
> 1) Number of parameters:
> Yes the latent dimension is indeed the same for G2G and the other compared methods. As mentioned in Sections 3.2 and 4.4 we always use **diagonal** covariance matrices which only have free parameters on the diagonal (and zeros everywhere else). Thus, an L-dimensional Gaussian with a diagonal covariance has L + L free parameters (mean + variance terms) and using only half of the competitors' dimensionality is a fair comparison.
>
> You are correct - in general, an L-dimensional Gaussian distribution has L+L(L-1)/2 free parameters, but only if we use a **full** covariance matrix. Our choice of diagonal covariances leads not only to fewer parameters but it also has computational advantages (e.g. it's easy to invert a diagonal matrix). We now highlight this choice one more time in the evaluation section for increased clarity.
>
> 2.1) Inductive learning:
> To see why the attributes play a role in the energy function notice that \mu_i and \sigma_i are not free parameters (i.e. to be updated by gradient descent), but they are rather the output of a parametric function that takes the node's attributes x_i as input. More specifically, as mentioned in Section 3.2 (and also in the appendix) \mu_i and \sigma_i are the outputs of a feed-forward neural network that takes the attributes of the node as input. During learning we do not directly update \mu_i and \sigma_i, but rather we update the weights of the neural network that produces \mu_i and \sigma_i as output.
>
> As mention in the discussion (Section 3.4) during learning we need both the attributes and the network structure (since the loss depends on the network structure). However, once the learning concludes, we essentially have a learned function f_theta(x_i) that only needs the attributes (x_i) of the node as input to produce the embedding (\mu_i and \sigma_i) as output. This is precisely what enables G2G to be inductive.
>
> 2.2) Raw attributes:
> We do indeed already compare the performance when using the raw attributes. The "Logistic Regression" method shown in Table 1 and 2, as well as Figures 1 and 2, is trained using only the raw attributes and it actually shows strong performance as a baseline. However, on the inductive learning task specifically, as we see in Table 2, G2G has significantly better performance compared to the logistic regression method that uses only the raw attributes.
>
> 3) Similarity/Dissimilarity:
> We agree with your comment w.r.t. similarity/dissimilarity and the KL divergence, this is essentially a typo and is already fixed in the uploaded revised version. This question was also asked earlier and we answer it in more details in the comment below.

---

> > ### Comment · AnonReviewer1 · 2017-12-04
> > **Review concerns are clarified; review rating updated.**
> >
> > The authors have clarified my questions, which are summarized as follows.
> > 1. The covariance matrices are actually assumed to be diagonal so the embedding vector length comparison is fair.
> > 2. How the raw attributes interact with the proposed network model are highlighted and explained.
> > 3. The Similarity/Dissimilarity issue is addressed.
> >
> > Therefore, I changed my rating from 5 to 7 due to the good quality and important impact of this work on node embedding.

---

### Public Comment · (anonymous) · 2017-11-23
**Comments on "Deep Gaussian Embedding of Graphs: Unsupervised Inductive Learning via Ranking"**

I have some problems about this paper:

1. In loss function, the authors employed the square-exponential loss.
   As shown in "A Tutorial on Energy-Based Learning", optimizing this loss function will make E_ij_k lower than E_ij_l.
   E_ij represents the opposite of the KL divergence.
   The smaller the KL divergence, the larger the similarity between the two distributions.
   This would make the similarity of i and j_k smaller than that of i and j_l, which is contrary to the previous assumption.
   Meanwhile, I tried to reproduce the experiment on the cora dataset and found that the loss fails to converge.

2. In problem definition, Sigma_i is a L*L matrix.
   But in experiment, it becomes a L-dimensional vector.
   Is the assumption of independence in each dimension reasonable?

3. In time complexity analysis, the authors ignore the complexity of a single iteration, which should be related to dimension L. Meanwhile, the authors do not explain the complexity of calculating the shortest path between nodes.

4. There is a notation error in the definition of KL divergence, that d should be changed to L.

---

> ### Author Response · Authors · 2017-11-24
> **Re: Comments on "Deep Gaussian Embedding of Graphs: Unsupervised Inductive Learning via Ranking"**
>
> Thank you very much for your interest in our paper and your comment.
>
> 1. You are right, this is a typo, which is an artifact of an earlier version of the paragraph where we used to talk about the negative energy instead. It should be E_{ij} = D_{KL}(N_j || N_i). This is indeed what we have implemented in our code. As you also can see in Section "3.3 Sampling strategy", we require E_{i1} < E_{i2}, ..., E_{iK-1} < E_{iK}, meaning nodes at a shorter distance should have lower energy/KL divergence.
>
> Notice that for calculating the performance in the link prediction task (e.g. area under the ROC curve) we indeed want to use -E_{ij} (the negative energy) as the score since two nodes should have a *higher* score if they are more likely to form an edge.
>
> Regarding the reproducibility comment, we are planning on releasing the code soon. We are also confident that by including the above correction (i.e. flipping the sign and using -E_ij for link prediction) you will be able to reproduce the results.
>
> 2. As demonstrated by the experiments, the assumption that the dimensions are independent/uncorrelated (to be precise, this assumption only applies for each single Gaussian, and not for all nodes jointly) performs well in practice. While it is straightforward to extend the model to full covariance matrices (e.g. using the Cholesky decomposition), this will lead to a significant computational overhead. Furthermore, previous approaches that learn Gaussian embeddings for other tasks (see our related work section) also have the same assumption and also show good performance in their experiments.
>
> 3. Since we are using a standard feed-forward architecture, the complexity of each iteration should be clear, thus we omitted it. We can add it in a revised version for completeness.
>
> Similarly, the complexity for computing truncated shortest path is well know, but we can add it to the appendix for completeness. We can efficiently calculate the truncated shortest path with sparse matrix operations. Thus, this complexity is O(K*E) where K is the maximum shortest path we are willing to consider and E is the number of edges. Since we used for all our experiments K<=3, this one time computation is essentially linear in the number of edges.
>
> 4. Thank you for pointing out the typo. We will fix in the next revision.

---

### Author Response · Authors · 2018-01-05
**Revision summary**

Based on the reviewers' comments we have made the following improvements to our paper:
* Clarified the use of KL divergence as a dissimilarity measure and negative energy for ranking candidate links
* Fixed several typos and improved wording in a few places

---

### Decision · Program_Chairs · 2018-01-29
**ICLR 2018 Conference Acceptance Decision**

**Decision:**

Accept (Poster)

**Comment:**

The paper proposes a method to embed graph nodes into a gaussian distribution rather than the standard latent vector embeddings. The reviewers concur that the method is interesting and the paper is well-written especially after the opportunity to update.